# Proprioceptive limit detectors contribute to sensorimotor control of the *Drosophila* leg

Brandon G. Pratt[1,3], Chris J. Dallmann [1], Grant M. Chou[1], Igor Siwanowicz[2], Sarah Walling-Bell[1], Andrew Cook[1], Anne Sustar[1], Anthony Azevedo [1] & John C. Tuthill [1] ✉

Many animals possess mechanosensory neurons that fire when a limb nears the limit of its physical range, but the function of these proprioceptive limit detectors remains poorly understood. Here, we investigate a class of proprioceptors on the *Drosophila* leg called hair plates. Using calcium imaging in behaving flies, we find that a hair plate on the fly coxa (CxHP8) detects the limits of anterior leg movement. By reconstructing CxHP8 axons in an electron microscopy dataset, we found that they are wired to excite posterior leg movement and inhibit anterior leg movement. Consistent with this connectivity, optogenetic activation of CxHP8 neurons elicited posterior postural reflexes, while silencing altered the swing-to-stance transition during walking. Finally, we use comprehensive reconstruction of peripheral morphology and downstream connectivity to predict the function of other hair plates distributed across the fly leg. Our results suggest that each hair plate is specialized to control specific sensorimotor reflexes that are matched to the joint limit it detects. They also illustrate the feasibility of predicting sensorimotor reflexes from a connectome with identified proprioceptive inputs and motor outputs.

Animals rely on proprioception to sense and coordinate movements of the body. One key function of the proprioceptive system is to detect when a body part has reached the limits of its normal range. In vertebrates, including mammals, the extremes of joint position are detected by low-threshold Ruffini endings and Pacinian corpuscles embedded within joint capsules[1]. In arthropods, including insects, joint limits may be detected by hair plates, fields of small, stiff mechanosensory hairs positioned at cuticular folds within joints[2]. Compared to proprioceptors that encode limb position and movement, such as vertebrate muscle spindles and invertebrate chordotonal organs, less is known about the physiology and behavioral function of proprioceptors that detect joint limits[3].

Early recordings from joint receptors in cats found that single afferents fired in a phasic or tonic manner near the limit of the range of joint movement[1]. For much of the 20th century, it was believed that these receptors were the primary source of our conscious sense of body position, or kinesthesia[4]. However, humans retain their kinesthetic sense after joint capsules are removed following hip replacement surgery[5]. This and other evidence[6] suggests that muscle spindles are the primary proprioceptors underlying kinesthesia in mammals. Compared to muscle spindles, far less is known about the downstream connectivity and function of vertebrate joint receptors[4].

In insects and other invertebrates, feedback from proprioceptive limit detectors has been shown to initiate stabilizing reflexes that sculpt motor patterns and protect the limbs from injury[7]. For example, classic work in the cockroach showed that hair plate neurons at a proximal leg joint monosynaptically excite extensor motor neurons and indirectly inhibit flexor motor neurons, restricting the range of joint flexion[8,9]. The ablation of this hair plate caused the corresponding leg to overstep and collide with the more anterior leg during walking, indicating that limit detection is important for controlling phase transitions of the step cycle[8]. Past work has identified specific sensorimotor reflexes mediated by insect hair plates, but it has been difficult to synthesize into a more comprehensive understanding of the

[1]Department of Neurobiology and Biophysics, University of Washington, Seattle, WA, USA. [2]Janelia Research Campus, Howard Hughes Medical Institute, Ashburn, VA, USA. [3]Present address: Allen Institute for Neural Dynamics, Seattle, WA, USA. ✉e-mail: tuthill@uw.edu

underlying feedback circuits and their function. Like many proprioceptors, hair plates are distributed throughout the animal's body, which raises the possibility that each sensory organ is specialized for sensing and controlling specific movements.

With emerging connectomic wiring maps[10–18] and genetic tools to target specific cell types[19,20], the fruit fly, *Drosophila*, has recently emerged as an important model system to investigate neural mechanisms of limb proprioception. A suite of techniques, including connectomics[14,21,22], electrophysiology[23], calcium imaging[24–28], genetic manipulations[29,30], and perturbations during behavior[28,31–38], have been applied to investigate the function of proprioceptors in the fly's femoral chordotonal organ (FeCO). The FeCO contains subtypes of mechanosensory neurons that detect the position, movement, and vibration of the fly femur-tibia joint, in a manner analogous to vertebrate muscle spindles[26,27,39]. However, compared to the FeCO, little is currently known about the anatomy, physiology, and behavioral function of hair plates on the *Drosophila* leg.

The six *Drosophila* legs have 214 hair plate mechanosensory neurons that are clustered into 42 hair plates (Supplementary Fig. S1A)[40–42]. Investigating the physiology and function of each hair plate, one by one, would be experimentally intractable, even in the tiny fruit fly. Instead, we start by studying one specific hair plate on the coxa of the fly's front legs (CxHP8) using in vivo calcium imaging (Fig. 1), connectomics (Fig. 2), and genetic manipulations during behavior (Figs. 3, 4). We find that the sensory tuning and motor reflex action of CxHP8 are well predicted by its external morphology and downstream connectivity to leg motor neurons. We therefore extend these sensorimotor reflex predictions to other hair plates positioned at different locations on the leg (Fig. 5). Overall, our results provide insight into the behavioral function of proprioceptive limit detectors.

## Results

### CxHP8 neurons detect the limits of coxa rotation and adduction

On the fly's front leg, three hair plates are located at the junction between the coxa (Cx) and thorax (Fig. 1A). These hair plates (CxHP8, CxHP3, CxHP4) wrap the joint along the anterior-posterior axis. We screened Split-GAL4 driver lines to identify lines that label coxa hair plates (Supplementary Table S1). The most specific and complete line we found[43] labeled CxHP8 neurons on the front and middle legs (6 and 3 out of 8 cells, respectively). Using this specific driver line, which we refer to as CxHP8-GAL4 (Fig. 1B and Supplementary Fig. S1B), we investigated the joint angle encoding properties of CxHP8 neurons.

We used a setup[25] that combines 2-photon calcium imaging with high-speed video to simultaneously record the activity of CxHP8 axons in the fly's ventral nerve cord (VNC) and 3D leg joint kinematics in CxHP8-GAL4 flies expressing GCaMP7f and tdTomato (Fig. 1C). To overcome the slow temporal dynamics of GCaMP7f relative to the dynamics of active leg movement in flies, we passively (and slowly) moved the left front leg using a manually controlled 3-axis platform (Supplementary Video 1). We then used DeepLabCut[44] and Anipose[45] to compute the 3 angles corresponding to the rotation, adduction, and flexion of the front leg thorax-coxa joint (Fig. 1D). Increases in rotation, adduction, and flexion angles correspond to the coxa rotating inward, translating medially, and flexing, respectively (Supplementary Fig. S1C). Our goal in these experiments was to determine the relationship between thorax-coxa joint angles and CxHP8 calcium activity (Fig. 1E).

Based on the anterior position of CxHP8 at the thorax-coxa joint, we hypothesized that its hairs would be deflected during anterior and medial movements, and inward rotations of the coxa (Supplementary Video 2). A representative trial illustrates that calcium activity peaked when the coxa was inwardly rotated and adducted (Fig. 1F). Across flies, we found similar tuning to coxa rotation and adduction (Fig. 1G). Although with greater variability, CxHP8 axons showed higher activity

when the coxa was extended. Because coxa rotation and adduction produced strong calcium responses, we next investigated how the two angles jointly influenced the activity of CxHP8 neurons. We found that the combination of both adduction and inward rotation resulted in maximal calcium activity of CxHP8 axons (Fig. 1H). Altogether, our results show that CxHP8 neurons detect the limits of anterior leg movement.

### CxHP8 neurons are active across behaviors

We next sought to determine the activity of CxHP8 neurons during different behaviors. We used a similar imaging setup[25] but replaced the platform with an air-supported ball and allowed the fly to spontaneously behave (Supplementary Video 3). We found that resting, walking, and grooming produced rotation and adduction coxa joint angles within the range that CxHP8 neurons were active (Fig. 1I). As illustrated by a representative trace, calcium activity increased during grooming and walking compared to adjacent resting states (Supplementary Fig. S1D). The CxHP8 calcium signal varied across different grooming bouts, likely a reflection of the variation in front leg kinematics across those bouts and the corresponding differences in the amplitude of hair plate deflection. Peak calcium activity was largest for front leg grooming bouts, likely because CxHP8 cuticular hairs are continuously deflected when the legs extend in front of the fly. Thresholding peak calcium activity (Supplementary Fig. S1E), we found CxHP8 neurons were active across behaviors (Fig. 1J), particularly during extreme adduction and rotation angles (Supplementary Fig. S1F).

### Connectomics predicts that CxHP8 neurons drive posterior leg movement

Using connectomics, we next reconstructed and analyzed hair plate sensorimotor circuits to predict the role of limit detection in leg motor control. Some hair plate axons were previously reconstructed in an electron microscopy (EM) dataset of a male adult nerve cord (MANC)[22]. However, specific hair plate axons could not be identified in MANC due to poor reconstruction quality. We therefore reconstructed hair plate axons within an EM dataset of the Female Adult Nerve Cord (FANC)[10,14]. We first used previous annotations[14] to identify and proofread all CxHP8 axons projecting into the left leg neuromere (Fig. 2A). We identified the peripheral location of each hair plate axon in the FANC connectome using a combination of genetic driver lines, axonal morphology, axonal projection patterns, and an x-ray holographic nanotomography dataset of the fly leg[42] (See Methods for details). For example, CxHP8 cells are identifiable in the X-ray dataset because they are the only hair plate axons that project into the VNC through the ventral prothoracic nerve (VProN).

The entire FANC dataset has been automatically segmented into neurons using convolutional neural networks[10], but many cells still required manual proofreading. We proofread and annotated cells downstream of CxHP8 neurons and found that CxHP8 axons allocate 75% of their output synapses (2279 ± 482 synapses per CxHP8 axon) to premotor (interneurons that synapse on motor neurons: 58%) and motor (17%) neurons. These percentages are significantly higher than that of exteroceptive leg sensory neurons, but similar to FeCO proprioceptors[21], suggesting that hair plates are primarily involved in leg motor control.

The 69 motor neurons controlling the fly's left front leg are organized into 14 motor modules – motor neurons within a module receive common presynaptic input and control the same joint[10]. Premotor neurons in FANC have also been annotated based on their developmental hemilineage[46], from which it is possible to infer the neurotransmitter released by each cell (acetylcholine, GABA, or glutamate). We first examined the connectivity from CxHP8 axons to premotor neurons to leg motor neurons (Fig. 2B). CxHP8 neurons provide strong monosynaptic input to motor neurons within the coxa

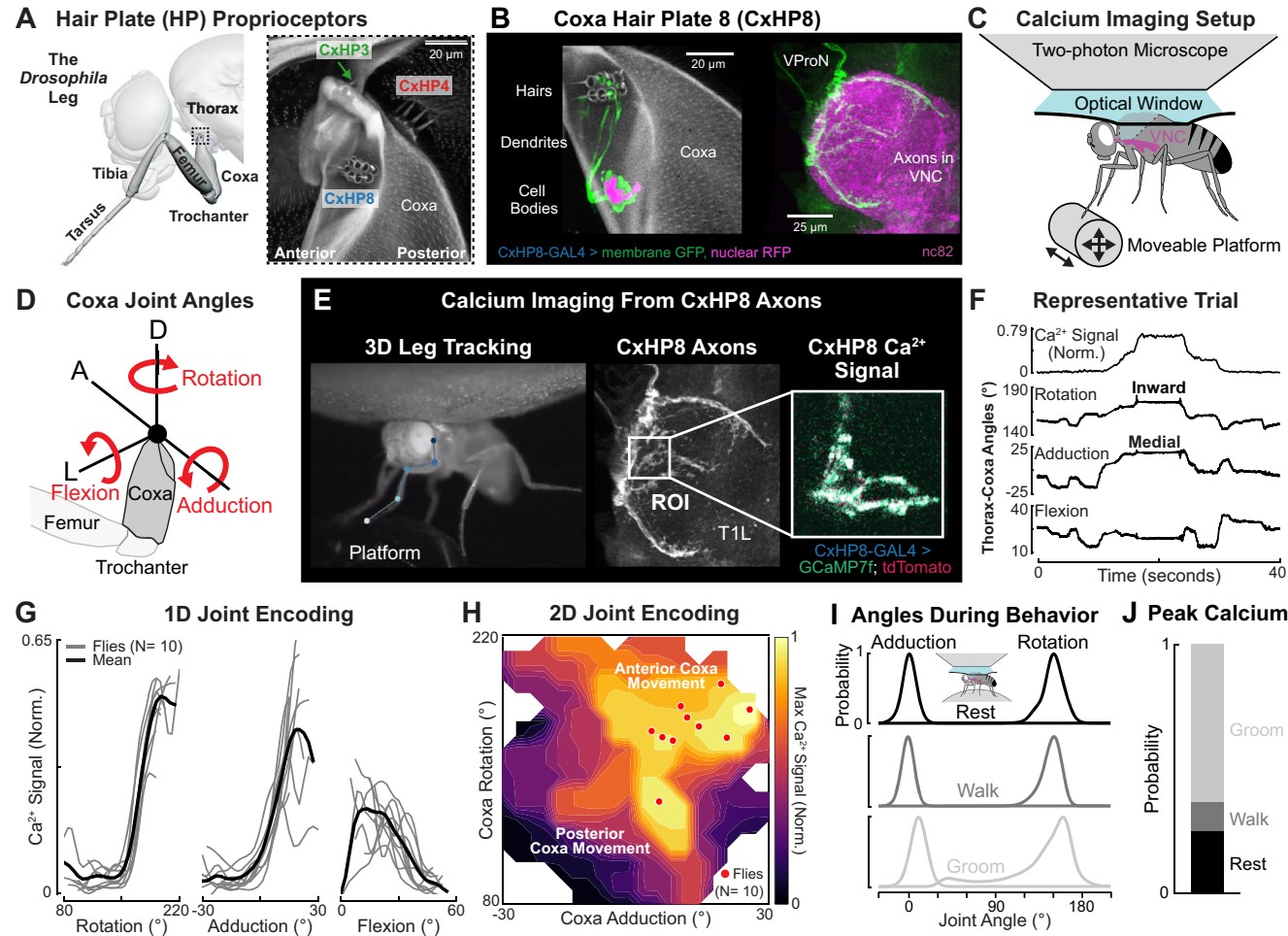

**Fig. 1 | CxHP8 neurons encode the anterior limits of thorax-coxa joint angles.**
**A** Left: schematic rendering of the fly's front leg. Right: a confocal image of the three hair plates, CxHP3, CxHP4, and CxHP8, at the thorax-coxa joint (black dotted box) of the front leg. **B** Confocal images of genetically labeled CxHP8 neurons (6 of 8) in the coxa of the front leg and ventral nerve cord (VNC). CxHP8 cuticular hairs: gray autofluorescence; CxHP8 neurons: GFP; Ventral prothoracic nerve (VProN): magenta (nc82); CxHP8 cell bodies: magenta (redstinger). The expression of CxHP8 in the front leg and VNC was imaged in three different flies. **C** Schematic of the setup used to measure calcium signals from CxHP8 axons as the left front leg was passively moved on a platform. **D** Schematic of the rotation, adduction, and flexion thorax-coxa joint angles. The red arrows start at 0 degrees and indicate positive changes in each joint angle. A: Anterior; D: Dorsal; L: Lateral. **E** Example recording of calcium activity of CxHP8 axons. **F** Representative trial showing the normalized calcium signal for CxHP8 axons, the thorax-coxa rotation, adduction, and flexion

angles while the left front leg was passively moved. Calcium activity was normalized between 0 and 1, where 1 represents the greatest calcium activity in the dataset. **G** Normalized calcium activity of CxHP8 axons with respect to thorax-coxa rotation, adduction, and flexion angles. Black line: bin averaged mean calcium activity across all flies; Gray lines: bin averaged mean calcium activity for each fly. **H** Contour plot showing the maximum 2D calcium activity across all flies with respect to thorax-coxa rotation and adduction. Red dots: the rotation and adduction angles at which the normalized calcium activity was maximum for each fly. **I** Adduction and rotation joint angle normalized probability densities across flies during rest (black), walking (gray), and front leg grooming (light gray). The inset shows a schematic of the calcium imaging setup used to determine CxHP8 activity during behavior. **J** Probability of flies grooming, walking, or resting during peak calcium activity. See also Supplementary Fig. S1 and Supplementary Videos S1-3.

posterior module (Fig. 2B,C). They also connect indirectly, via excitatory and inhibitory premotor neurons, to multiple other motor modules across the leg (Fig. 2C and Supplementary Fig. S2A). CxHP8's strongest inhibitory connections are onto the coxa promote module, an antagonist of the coxa posterior module (Fig. 2B). We also found extensive recurrent connectivity within and across premotor neurons, as well as between CxHP8 axons and 19A premotor neurons (Supplementary Fig. 2B). Overall, CxHP8 neurons are positioned to provide feedback to multiple motor modules, but with the strongest excitation to motor neurons driving posterior leg movement and inhibition to motor neurons driving anterior movement. These connectivity patterns lead us to hypothesize that activity in CxHP8 neurons drives posterior leg movement when the coxa reaches its anterior limit. For example, CxHP8 feedback could contribute to the swing-to-stance transition during walking (Fig. 2E).

## CxHP8 activation drives posterior leg movement and silencing alters swing-to-stance transitions during walking

We next tested whether CxHP8 activation contributes to posterior leg movements, as predicted by their synaptic connectivity to leg motor neurons. As in prior studies[23,47], we used a setup that enables spatiotemporally precise optogenetic activation or silencing of neurons in tethered flies behaving on an air-supported ball (Fig. 3A). Using six high-speed cameras, we reconstructed full 3D leg joint kinematics using DeepLabCut[44] and Anipose[45]. To optogenetically activate CxHP8 neurons, we genetically expressed ChrimsonR within the CxHP8 neurons and illuminated the thorax-coxa joint of the left front leg with a red laser. In a representative standing fly, we observed that its left front leg moved posteriorly shortly after the onset of the laser, which was accompanied by outward rotation, lateral movement, and flexion of the coxa (Fig. 3B, C and Supplementary Video 4). We observed the

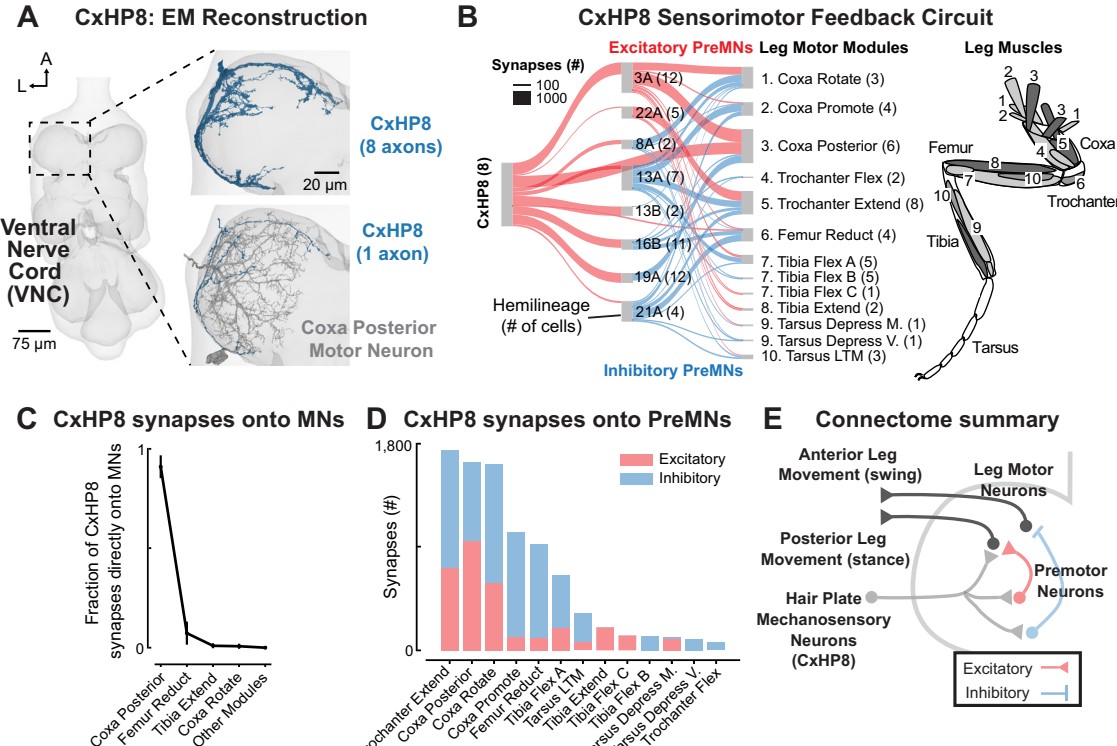

**Fig. 2 | Synaptic connectivity suggests that CxHP8 neurons connect to a reflex circuit that drives posterior leg movement. A** We reconstructed all 8 CxHP8 axons within one left front leg neuromere of a *Drosophila* female VNC (FANC) electron microscopy volume. **B** CxHP8 axons synapse onto premotor neurons, identified by their hemilineage, as well as motor neurons grouped into motor modules[10]. Red and blue lines signify excitatory and inhibitory connections, respectively. The strength of the connection (width of the line) is the number of synapses all neurons within a cell class make onto a downstream cell class. Only premotor and motor neurons that received an average of 4 or greater synapses per neuron from an upstream cell class were analyzed (i.e., CxHP8 input threshold: 8 cells * 4 synapses). The number of cells in each cell class included in this analysis is indicated within parentheses. The numbers in the leg schematic on the right illustrate the muscles belonging to leg motor modules 1–10. **C** CxHP8 neurons primarily synapse onto motor neurons within the coxa posterior motor module. Error bars are the standard deviation of the mean fraction of motor module input by the 8 CxHP8 neurons ($n = 8$). **D** Relative number of excitatory and inhibitory synapses between premotor neurons that receive input from CxHP8 neurons and motor neurons within motor modules. Red and blue bars indicate excitatory and inhibitory connections, respectively. **E** Summary schematic showing the circuit through which CxHP8 neurons are predicted to drive posterior leg movement. See also Supplementary Fig. S2.

same changes in thorax-coxa joint angles across standing flies when CxHP8 neurons were optogenetically activated (Fig. 3D), whereas control flies lacking Chrimson showed negligible kinematic changes during laser stimulation (Supplementary Fig. S3A). These joint angle changes were accompanied by posterior, lateral, and downward displacement of the distal tarsus (Fig. 3E). Overall, the activation of CxHP8 neurons produced posterior and lateral movement of the stimulated leg, without impacting the other legs.

We next optogenetically silenced CxHP8 neurons through the expression of a light-gated chloride channel (GtACR1) during spontaneous behaviors. When CxHP8 feedback was silenced during forward walking, the thorax-coxa adduction angle of the left front leg was slightly but significantly more medial, particularly at the swing-to-stance transition of the step cycle (Fig. 3F). We did not observe medial coxa displacements in control flies that experienced the same laser stimulation (Supplementary Fig. S3B). Optogenetic silencing of CxHP8 neurons also subtly but significantly altered the tarsus position at the swing-to-stance transition (Fig. 3G). Thus, the lack of CxHP8 activity appeared to cause the coxa to slightly but significantly overshoot the normal kinematic range associated with the swing-to-stance transition, leading to a displacement of the entire leg. We also assessed the role of CxHP8 feedback in left (Supplementary Fig. S3C) and right turns (Supplementary Fig. S3D), and front leg grooming (Supplementary Fig. S3E). Although joint angles were not affected during left turns, we found that the coxa was significantly more extended and rotated inward at the swing-to-stance transition when CxHP8 neurons were silenced during right turns. An explanation for this asymmetry is that the left leg moves inward during right turns (but not left turns), which would deflect the hairs of CxHP8 to a greater extent, given their anterior, inward position within the thorax-coxa joint. The coxa was also subtly but significantly more rotated, adducted, and extended in flies with CxHP8 neurons silenced during front leg grooming (Supplementary Fig. S3E).

## CxHP8 facilitates the swing-to-stance transition and controls resting posture in untethered flies

Although convenient for delivering targeted optogenetic stimuli and tracking joint kinematics, flies walking in the tethered configuration have slightly different walking kinematics than untethered flies[35]. To test whether CxHP8 feedback is important for the control of swing-to-stance transitions in untethered walking flies, we used a linear treadmill system[35]. We genetically expressed an inward rectifying potassium channel, Kir 2.1, to chronically hyperpolarize CxHP8 neurons. As flies walked on the treadmill, driven at a belt speed of 10 mm/s, we recorded their movement at 200 fps with 5 high-speed cameras (Fig. 4A). From the high-speed videos, we reconstructed the 3D positions of the head, thorax, abdomen, and each leg tip using DeepLabCut[44] and Anipose[45]. We then used these reconstructed 3D positions to quantify kinematics and posture during walking and rest periods, respectively (see Methods for details; Fig. 4B).

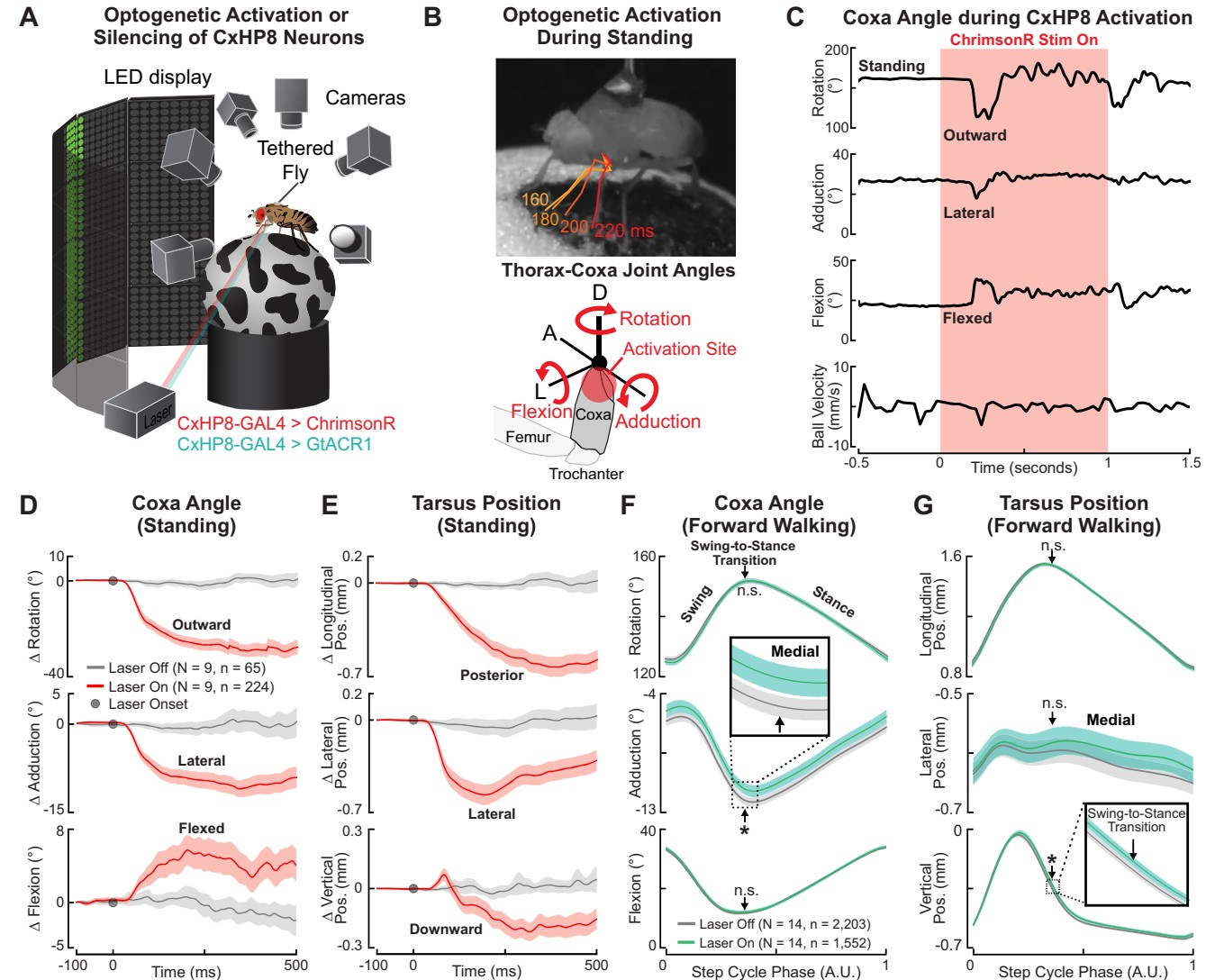

**Fig. 3 | CxHP8 neurons drive posterior leg movement to mediate the swing-to-stance transition during walking. A** Schematic of the optogenetic activation and silencing setup. **B** Top: example trial in which CxHP8 activation resulted in outward rotation, lateral movement, and flexion of the coxa in a standing fly after laser onset (in ms). Bottom: the rotation, adduction, and flexion thorax-coxa joint angles. Red arrows: 0 degrees to positive joint angles; A: Anterior; D: Dorsal; L: Lateral. **C** For the same trial shown in (**B**), the thorax-coxa rotation, adduction, and flexion angles, and the ball velocity in response to the optogenetic activation of CxHP8 neurons. **D** Optogenetic activation (red) of CxHP8 neurons in standing flies resulted in significant outward rotation, lateral displacement, and flexion of the coxa compared to laser off trials (gray). The joint angle trajectories are normalized to the start of the analysis window (i.e., 100 ms prior to laser onset). Shaded region: 95% confidence interval of the mean; N: number of flies; n: number of trials. **E** Optogenetic activation (red) of CxHP8 neurons in standing flies also resulted in significant

lateral, downward, and posterior movements of the tarsus compared to laser off trials. Shaded region: 95% confidence interval of the mean; N: number of flies; n: number of trials. **F** Optogenetic silencing (green) of CxHP8 neurons in flies walking forward produced a significant (*: $p < 0.05$; $p = 0.019$) medial movement of the coxa at the swing-to-stance transition compared to laser off trials (gray). Arrow: average swing-to-stance transition of the step cycle; Shaded region: 95% confidence interval of the mean; N: number of flies; n: number of steps. A linear mixed-effects model was used to statistically compare the distributions of joint angles at the swing-to-stance transition. **G** Optogenetic silencing (green) of CxHP8 neurons in flies resulted in a significantly displaced (*: $p < 0.05$; $p = 0.01$) tarsus position compared to laser off trials (gray). A linear mixed-effects model was used to statistically compare the distributions of tarsus positions at the swing-to-stance transition. Shaded region: 95% confidence interval of the mean. See also Supplementary Fig. S3 and Supplementary Video 4.

Like tethered flies, we found that silencing CxHP8 neurons in flies walking on the treadmill impaired step kinematics of the front legs near the swing-to-stance transition. In particular, the anterior extreme position (AEP), where a leg first enters stance, was displaced medially and posteriorly compared to controls (Fig. 4C). The posterior extreme position (PEP), where a leg first enters swing, was also significantly displaced in a similar manner to the AEP. Given that the AEP marks the swing-to-stance transition, and that changes in the AEP may drive changes in the PEP, we focused on understanding the kinematic changes that facilitate the displacement of the AEP in CxHP8-silenced

flies. In examining the average swing trajectories of the front legs, we found a significant medial deviation of the trajectory near the swing-to-stance transition in CxHP8 silenced flies compared to controls (Fig. 4D). Although this deviation did not result from a significantly longer mean swing duration (Fig. 4E), the mean swing distance was significantly greater in flies with CxHP8 neurons silenced (Fig. 4F). Unlike front leg step kinematics, inter-leg coordination was not significantly altered in flies with CxHP8 neurons silenced (Fig. 4G). Overall, the silencing of CxHP8 neurons in flies walking on the treadmill resulted in more medial front leg movement near the swing-to-

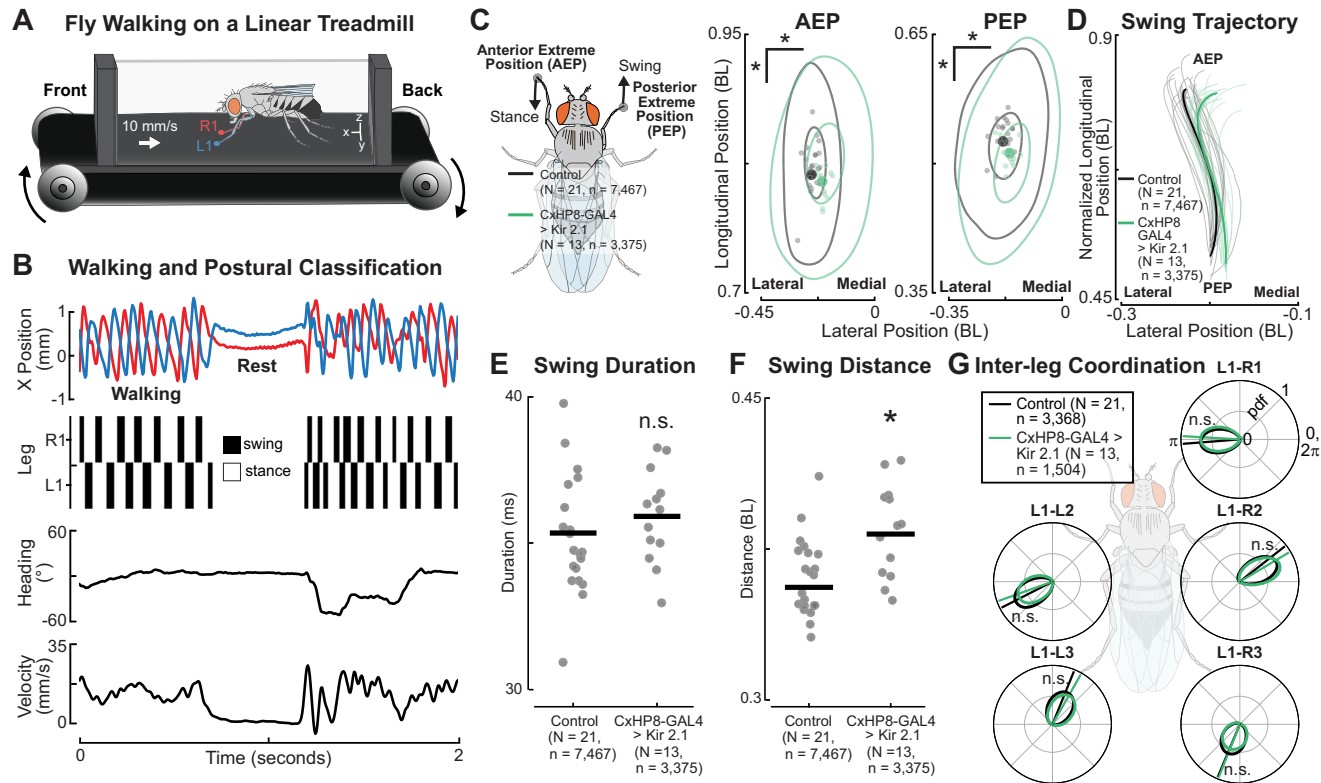

**Fig. 4 | CxHP8 neurons control swing-to-stance transitions in unconstrained flies. A** Schematic of the treadmill system. **B** Example trial period illustrating that both walking kinematics and resting posture can be obtained using the treadmill system. Shown are the longitudinal (X) position and the classification of swing and stance of the left and right front leg as the fly walked and rested on the treadmill belt. **C** Left: Schematic of the anterior and posterior extreme positions. Right: Kernel density estimation plots of the anterior and posterior extreme positions significantly shifted medially and posteriorly in flies with CxHP8 neurons chronically silenced (green) with the inward rectifying potassium channel, Kir 2.1. N: number of flies; n: number of steps. A linear mixed-effects model (see Methods for details) was used to statistically compare CxHP8 silenced and control flies (*: $p < 0.05$; AEP Longitudinal: $p = 5.96e^{-10}$; AEP Lateral: $p = 5.87e^{-210}$; PEP Longitudinal: $p = 3.95e^{-21}$; PEP Lateral: $p = 1.24e^{-52}$). **D** Swing trajectories of CxHP8 silenced (green) and control (black) flies demonstrate the medial overshoot near the swing-to-

stance transition. Although kinematic results were largely consistent between tethered and untethered flies, the tarsus position was not significantly displaced medially in untethered flies when CxHP8 neurons were silenced. This difference could be a consequence of the difference in substrate[35], tethering, or silencing method.

In larger insects, hair plates have been shown to be important for controlling resting body[48] and leg posture[7,49]. We therefore examined the posture of flies during rest periods on the treadmill. Even though flies with CxHP8 neurons silenced did not have significantly different body heights (Supplementary Fig. S4A) or angles (Supplementary Fig. S4B), the resting positions of their tarsi were significantly more splayed out than controls (Supplementary Fig. S4C). We further quantified this by calculating the area of the polygon formed by the tarsi positions (Supplementary Fig. S4D). The mean polygon area of CxHP8 silenced flies was significantly greater than controls, providing further support that CxHP8 neurons control the resting posture of the legs.

### Connectome analysis enables functional predictions for other hair plates on the fly leg
Our calcium imaging data indicate that CxHP8 neurons function as limit detectors of anterior coxa movement (Fig. 1). Based on analysis of

stance transition when CxHP8 neurons were silenced. Each trajectory was bounded between the average PEP and AEP for each fly and across flies. Solid line: mean trajectory; Thin lines: mean trajectory of each fly; Shaded region: 95% confidence interval; N: number of flies; n: number of steps. **E** Swing duration was not statistically different between CxHP8 silenced and control flies, based on a two-sided $t$ test with a Bonferroni correction of 2 on the means of flies (n.s.: $p > 0.025$). **F** Swing distance was significantly greater for CxHP8 silenced flies compared to controls (*: $p < 0.025$; $p = 0.006$) based on a two-sided $t$ test with a Bonferroni correction of 2 on the means of flies. **G** Inter-leg coordination (relative step cycle phase) between the left front leg and each other leg was not significantly different between CxHP8-silenced and control flies. A two-sided Kuiper two-sample test with a Bonferroni correction of 5 determined significance (n.s.: $p > 0.01$). N: number of flies; n: number of phase comparisons. See also Supplementary Fig. S4.

the connectome, we hypothesized that feedback from CxHP8 neurons helps transition the coxa away from its anterior limit by driving posterior movement (Fig. 2). We tested and confirmed this prediction by genetically manipulating the activity of CxHP8 neurons in behaving flies (Figs. 3, 4). Having validated the predicted reflex function for one hair plate, we now return to the connectome to predict the role of other front leg hair plates in leg motor control.

We focused on the two other hair plates on the front leg coxa, CxHP3 and CxHP4, along with the three hair plates located at the coxa-trochanter joint, TrHP5, TrHP6, and TrHP7 (Fig. 5A, Supplementary Fig. S1A and Supplementary Video 2). In FANC, we reconstructed the axons of each of these hair plates (Fig. 5B and Supplementary Fig. S5A). Of the thousands of output synapses from these hair plate axons (Supplementary Fig. S5B), the majority of synapses are onto local premotor and motor neurons (Fig. 5C). Interestingly, we also found that some hair plate axons synapse onto glia (Fig. 5C and Supplementary Fig. S5C), suggesting that their signaling may be used for non-motor related functions, such as detecting injury. The axons of each hair plate connect to distinct populations of motor neurons that actuate the leg segment the hair plate is located on (Fig. 5D). For instance, CxHP4 axons provide strong connectivity to motor neurons of the coxa rotate motor module, which drives anterior coxa

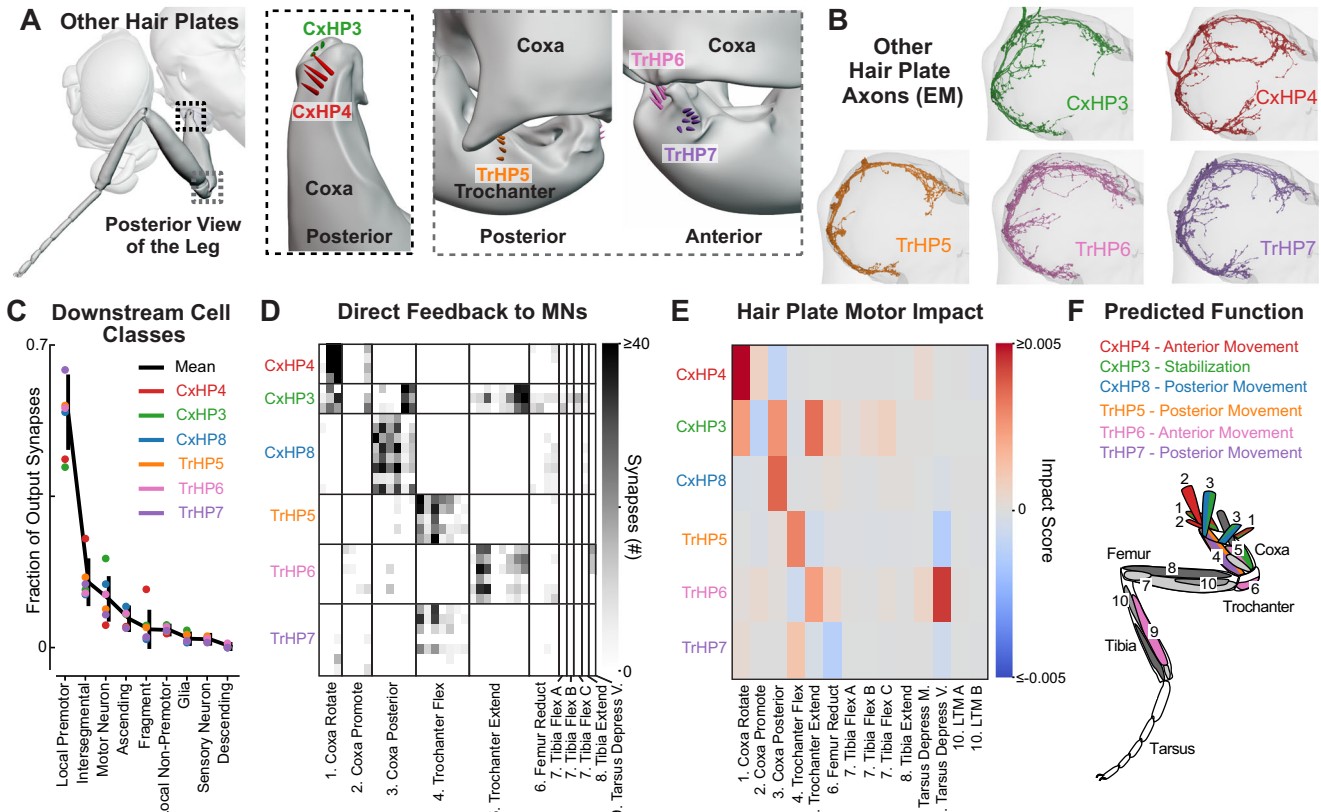

**Fig. 5 | Synaptic connectivity of hair plate axons indicates distinct roles in leg motor control. A** Hair plates located at the thorax-coxa (black dotted box) and coxa-trochanter (gray dotted box) joints of the front leg. Schematics were rendered in Blender and based on high-resolution confocal images. We focused on two hair-plates, in addition to CxHP8, located at the thorax-coxa joint (CxHP4: red; CxHP3: green) and three at the coxa-trochanter joint (TrHP5: orange; TrHP7: purple; TrHP6: pink). Posterior and anterior refer to the face of the joint that is being viewed. See Supplementary Video 2 for an animation of the locations of the hair plates on the front leg (**B**) Reconstructed hair plate axons within the left front leg neuromere of FANC. **C** Fraction of hair plate axon output synapses onto VNC cell classes. Black line: mean fraction of output synapses across all hair plate axons (*n* = 33); Error bars: standard deviation; Color dot: mean fraction of output synapses for the axons of each hair plate. **D** Connectivity matrix shows the number of synapses that motor neurons receive from hair plate axons. Axons from the same hair plate are grouped together (rows) while motor neurons within their defined motor module are arranged according to the proportion of input synapses they receive from specific hair plate axons. The leg muscles corresponding to the motor modules 1–9 are shown in (**F**). **E** Motor impact scores[21], weighted sums of direct (monosynaptic) and indirect (di-synaptic) signed connection strengths, of hair plates onto leg motor modules. Red and blue hues correspond to excitatory and inhibitory impact scores, respectively. The leg muscles corresponding to motor modules 1–10 are shown in (**F**). See Methods for more information on how the motor impact score was calculated. **F** Predicted function and muscles controlled by each hair plate, as indicated by corresponding colors. See also Supplementary Fig. S5 and Supplementary Video 2.

movement, whereas the antagonistically positioned CxHP8 connects to the antagonistic coxa posterior motor module. We also found that the axons of each hair plate synapse onto diverse premotor neurons (Supplementary Fig. S5D). By combining the connectivity of hair plates onto premotor and motor neurons, we constructed 2-layer sensorimotor circuits for each hair plate (Supplementary Fig. S5E). We also computed the motor impact score for each hair plate (Fig. 5E), a measure that weights direct and indirect connectivity between sensory and motor neurons, incorporating the neurotransmitter of any intervening interneurons[21]. This impact score is useful to understand trends or differences in motor connectivity, but, importantly, does not consider factors such as activity dynamics or intrinsic neural properties. Nonetheless, comparing impact scores revealed clear differences in motor connectivity among hair plates on different parts of the leg. For example, our analysis predicts that CxHP4 drives anterior leg movement, CxHP3 reinforces both anterior and posterior leg movement, TrHP5 and TrHP7 drive posterior leg movement, and TrHP6 drives anterior leg movement. Overall, the connectivity of hair plates suggests each limit detector is specialized for controlling specific leg movements that are matched to the different extremes of joint movement that they detect (Fig. 5F).

## Discussion

In this study, we analyzed the physiology, circuit connectivity, and behavioral function of hair plates on the fly leg. We first used calcium imaging to discover that the neurons of one hair plate (CxHP8) are tonically active at the anterior extremes of coxa movement (Fig. 1). We then used connectomics to predict that feedback from CxHP8 neurons drives posterior coxa and leg movement (Fig. 2). Optogenetic stimulation of CxHP8 neurons was sufficient to drive reflexive posterior leg movements (Fig. 3). We also found that optogenetic and chronic silencing of CxHP8 neurons in tethered flies (Fig. 3) and flies walking freely on an actuated treadmill (Fig. 4), caused the leg to overshoot the swing-to-stance transition in a medial direction. In addition to their role in walking, silencing CxHP8 neurons altered the resting position of the legs (Supplementary Fig. S4). Overall, these results agree with the sensorimotor reflex predictions from the connectome. We also provide functional predictions for the other hair plates on the front leg (Fig. 5), which can be tested in future experiments as new genetic driver lines become available. Altogether, this work illustrates an integrated approach, anchored in the connectome, to infer the sensorimotor functions of spatially distributed proprioceptors.

Proprioceptive limit detectors are common in limbed animals. Nearly fifty years ago, recordings from joint receptors in the cat leg found that single afferents fired in a phasic or tonic manner as they neared the limit of the range of joint movement[1]. Previous electrophysiological work in locusts[50–54] and cockroaches[8,9,55,56] also found neurons within the same hair plate had either tonic or phasic encoding properties. However, we found no evidence for phasic tuning among CxHP8 axons – calcium signals remained sustained when the leg was held at an extreme position. It is possible that all hair plate neurons in the fly have tonic encoding properties. It is also possible that other hair plate neurons, which we did not record from in this study, respond phasically to mechanical stimulation. More work is needed to determine whether physiological properties vary across hair plate neurons.

While extensive prior work has investigated the encoding properties of proprioceptive limit detectors in other animals, less was known about their downstream connectivity and contributions to leg motor control during behavior. We found that activating and silencing CxHP8 neurons confirmed our leg motor control predictions based on connectivity. It is worth noting that the VNC circuits connecting hair plates and leg motor neurons are far more complex than our behavioral results and past work might suggest. For example, CxHP8 neurons also disynaptically excite and/or inhibit motor neurons controlling every other segment within the leg (Fig. 2B), suggesting leg-wide rather than leg segment-specific feedback control. Premotor neurons downstream of hair plates are also recurrently connected in complex patterns, and some synapse back onto CxHP8 axons (Supplementary Fig. S2B). This dense connectivity may support inter-joint coordination or provide robustness to external perturbations that the animal faces while navigating complex terrain.

Reflex circuits may also be modulated (i.e., suppressed or amplified) during different behavioral contexts. For example, hook (but not claw) proprioceptors in the fly femoral chordotonal organ (FeCO) are presynaptically inhibited during self-generated leg movements[25]. Here, we did not observe suppression of hair plate calcium activity during spontaneous leg movements (Fig. 1). Interestingly, optogenetic silencing of CxHP8 neurons had a significant, but relatively small, effect on extreme anterior extension of the coxa during front leg grooming (Supplementary Fig. S3E), even though CxHP8 neurons showed strong activity during this behavior (Fig. 1J, Supplementary Fig. S1D and Supplementary Video 3). Gain modulation could occur in downstream premotor neurons to fine-tune the reflex action of hair plates.

The fruit fly's front leg contains over 200 proprioceptive sensory neurons, including hair plates, campaniform sensilla, and chordotonal neurons[42]. Chordotonal neurons are primarily concentrated in the FeCO, which senses the femur/tibia joint[27]. The more proximal coxa and trochanter joints contain similar numbers of campaniform sensilla and hair plates. Connectomic and electrophysiological evidence suggest that signals from these different proprioceptor classes are highly integrated in downstream circuits. For instance, work in the stick insect has shown that load and movement signals encoded by campaniform sensilla and FeCO neurons, respectively, are integrated in premotor networks to control the femur-tibia joint[57]. Similarly, in the fly, electrophysiology and calcium imaging has revealed that signals from functional subtypes of FeCO neurons (i.e., claw, hook, club) are integrated in downstream neurons[23,24]. Connectomic reconstruction has also revealed convergence of claw, hook, hair plate, and campaniform sensilla in downstream VNC neurons[21]. In a prior study, we showed that chronically silencing a small subset of campaniform sensilla and hair plate neurons, including CxHP8, significantly impaired step kinematics and inter-leg coordination, especially at fast walking speeds[35]. Altogether, integration of signals across diverse proprioceptors likely contributes to locomotor robustness and could also help explain why silencing CxHP8 neurons had only a very subtle effect on joint kinematics during walking.

Here, we focused on the local reflex action of hair plates within a single leg. Hair plate axons do not project intersegmentally, and the majority of their disynaptic connectivity is also onto motor neurons controlling the same leg. However, past work has shown that hair plates may also contribute to intersegmental coordination across legs. In stick insects, for example, the ablation of hair plates on the middle leg resulted in non-linear and greater shifts in the anterior extreme position of the ipsilateral hind leg[58]. We did not observe kinematic changes of other legs when manipulating CxHP8 neurons on the fly's left front leg. However, intersegmental effects could be driven by other hair plates.

Unlike other senses with a dedicated and localized organ like an eye, ear, or nose, proprioception is a distributed sense that relies on mechanosensory neurons distributed throughout the body. Therefore, a fundamental challenge for understanding proprioception is that the same class of sensory neurons in different locations may be anatomically specialized to detect specific perturbations to the body. Investigating hundreds or thousands of proprioceptors, one-by-one, is experimentally intractable. A particular challenge is that there do not yet exist genetic methods in the fly to systematically label and manipulate individual proprioceptive organs. Indeed, despite an extensive search, we have not yet identified a genetic means to label all hair plates, or even all sensory neurons within one hair plate. A promising future approach to overcome this problem is to combine the methods used above (connectomics, calcium imaging, and genetic manipulations during behavior) with biomechanical modeling[59,60]. To facilitate this effort, we have added all of the hair plates characterized in this study to the open-source Janelia fly body Blender model[60] (Supplementary Video 2). As other components are added to the body model, it will become possible to simulate hair plate signaling through recurrent, connectome-constrained networks. In the future, integrating biomechanical and connectome-constrained network models may provide a promising solution to the challenge of understanding how distributed sensory systems, including proprioception, control complex behaviors.

## Methods

### Fly husbandry and genotypes
Experiments were performed on adult male flies, *Drosophila melanogaster*, between 2-5 days post-eclosion. Flies were reared in a 25 °C incubator with 14:10 light:dark cycle within vials filled with a standard cornmeal and molasses medium. Genotypes used in experiments are listed in Table 1.

### Blender model of *Drosophila* body and front leg hair plates
A morphologically accurate model of an adult *Drosophila melanogaster* containing the hair plates at the thorax-coxa and coxa-trochanter joints of the front leg was constructed using the 3D rendering and modeling software, Blender[60]. The body of the fly, and the anatomy and location of hair plates in the model, were based on high-resolution confocal images. We replayed the 3D walking kinematics of the left front leg coxa measured from a tethered fly walking on the ball to this model to illustrate the phases during the step cycle when the hairs of each coxa hair plate would be deflected (Supplementary Video 2).

### Confocal imaging of proprioceptor expression
**Peripheral expression of proprioceptors in legs.** Front, middle, and/ or hind legs of male flies associated with the driver lines in Supplementary Table S1 crossed with the UAS-mcd8GFP; redstinger reporter line were fixed in a 4% formaldehyde (PFA) PBS solution for 20 min and then washed three times in a 0.2% Triton X-100 PBS solution. Legs were then cleared with FocusClear (CelExplorer) and mounted on slides in MountClear (CelExplorer). The expression at each hair plate was assessed using an Olympus FV1000 confocal microscope. The number

**Table. 1 | *Drosophila melanogaster* genotypes used for experiments**

| Stock Name | Genotype | Figure(s) | Stock Source |
|---|---|---|---|
| R48A07 AD: R20C06 DBD > mCD8GFP; Redstinger | w[1118]/ yw, hs flp, UAS-mCD8-GFP; P{+t[7.7]w[=mC]=R48A07-p65.AD}attP40/ UAS-stingerRed; P{y[+t7.7]w[+mC]R20C06-GAL4.DBD}attP2/+ | 1, S1 | UAS-mCD8GFP; Redstinger was gifted from the Parrish lab, University of Washington and crossed with split-GAL4 gifted from Moon Lab, Yonsei University (Bloomington Stocks #69841 and #71070) |
| R48A07 AD: R20C06 DBD > pJFRC7-20XUAS-IVS-mCD8::GFP | w[1118]/w[*]; P{+t[7.7]w[=mC]=R48A07-p65.AD}attP40/ + ; P{y[+t7.7]w[+mC]=R20C06-GAL4.DBD}attP2/P{pJFRC7-020XUAS-IVS-mCD8::GFP}attP2 | 1, S1 | 20X UAS-mCD8::GFP was gifted from the Rubin lab and crossed with split-GAL4 gifted from Moon Lab, Yonsei University (Bloomington Stocks #69841 and #71070) |
| R48A07 AD: R20C06 DBD > GcaMP7f; tdTomato | w[1118]/ + DL; P{+t[7.7]w[=mC]= R48A07-p65.AD}attP40/ P{w[+mC] = UAS-tdTom.S}2; P{y[+t7.7]w[+mC]=R20C06-GAL4.DBD}attP2/ PBac{y[+mDint2]w[+mC]=20xUAS-IVS-jGCaMP7f}VK00005 | 1, S1 | UAS-GCaMP7f; tdTomato were contrused from Bloomington stocks #36327 and #79031, and crossed with split-GAL4 gifted from Moon Lab, Yonsei University (Bloomington Stocks #69841 and #71070) |
| R48A07 AD: R20C06 DBD > ChrimsonR (CxHP8 optogenetic activation) | w[1118]/ + DL; P{+t[7.7]w[=mC] = R48A07-p65.AD}attP40/+ DL; P{y[+t7.7]w[+mC] = R20C06-GAL4.DBD}attP2/10X UAS ChrimsonR mCherry (attp2)/TM3 Sb | 3 | UAS-ChrimsonR was gifted from Janelia (outcrossing was done by Anne Sustar, University of Washington) and crossed with split-GAL4 gifted from Moon Lab, Yonsei University (Bloomington Stocks #69841 and #71070) |
| R48A07 AD: R20C06 DBD > GtACR1 (CxHP8 optogenetic silencing) | w[1118]/ + DL; P{+t[7.7]w[=mC] = R48A07-p65.AD}attP40/+ DL; P{y[+t7.7]w[+mC]=R20C06-GAL4.DBD}attP2/20X-UAS-GtACR1-EYFP (III) attp2 | 3, S3 | UAS-GtACR1 stock was created by Anne Sustar in the Tuthill Lab at the University of Washington and was crossed with split-GAL4 gifted from Moon Lab, Yonsei University (Bloomington Stocks #69841 and #71070) |
| R52A01 DBD > GtACR1 (Inactivation control) | w[1118]/ + DL; +/+ DL; P{y[+t7.7]w[+ mC] = R52A01-GAL4.DBD}attP2/20X-UAS-GtACR1-EYFP (III) attp2 | S3 | UAS-GtACR1 stock was created by Anne Sustar in the Tuthill Lab at the University of Washington and was crossed with Bloomington Stock #69141 |
| R52A01 DBD > ChrimsonR (Activation control) | w[1118]/ + DL; +/+ DL; P{y[+t7.7]w[+ mC] = R52A01-GAL4.DBD}attP2/10X UAS ChrimsonR mCherry (attp2)/TM3 Sb | S3 | UAS-ChrimsonR was gifted from Janelia (outcrossing was done by Anne Sustar, University of Washington) and crossed with Bloomington Stock #69141 |
| R48A07 AD: R20C06 DBD > Kir 2.1 (CxHP8 chronic silencing) | w[1118]/ + DL; P{+t[7.7]w[= mC] = R48A07-p65.AD}attP40/+ DL; P{y[+t7.7]w[+ mC] = R20C06-GAL4.DBD}attP2/ pJFRC49-10XUAS-IVS-eGFP::Kir2.1(attP2) | 4, S4 | Crossed Dickinson Lab Stock U-111 with split-GAL4 gifted from Moon Lab, Yonsei University (Bloomington Stocks #69841 and #71070) |
| R48A07 AD > Kir 2.1 (Control) | w[1118]/ + DL; P{+t[7.7]w[= mC] = R48A07-p65.AD}attP40/+ DL; pJFRC49-10XUAS-IVS-eGFP::Kir2.1(attP2)/+ | 4, S4 | Crossed Dickinson Lab Stock U-111 with Bloomington Stock #71070 |
| R39B11 AD > Kir 2.1 (Control) | w[1118]/ + DL; P{+t[7.7]w[= mC]=R39B11-p65.AD}attP40/+ DL; pJFRC49-10XUAS-IVS-eGFP::Kir2.1(attP2)/+ | 4, S4 | Crossed Dickinson Lab Stock U-111 with Bloomington Stock #71040 |

of proprioceptor cells labeled by each driver line was then determined through visual inspection of the volumetric imaging stacks in FIJI[61].

**Central expression of proprioceptors in the brain and VNC.** The brain and VNC were dissected from male flies associated with the split-GAL4 driver lines that showed interesting peripheral expression in hair plates, such as the CxHP8 split-GAL4 driver line (i.e., R48A07 AD; R20C06 DBD). Unlike above, the GAL4 driver resulted in the expression of pJFRC7-20XUAS-IVS-mCD8::GFP. The brains and VNCs were fixed in the same manner as legs, but then were put into a blocking solution (5% goat serum, PBS, 0.2% Triton-X) for 20 min followed by being incubated in a blocking and primary antibody solution (1:50 concentration of anti-GFP chicken antibody, 1:50 concentration of anti-brp mouse) for 24 h at room temperature. The brains and VNCs were then washed three times in PBS with 0.2% Triton-X and incubated in a blocking and secondary antibody solution (1:250 concentration of anti-chicken-Alexa 488, 1:250 concentration of anti-mouse-Alexa 633). After this, the brains and VNCs were washed three times with PBST and mounted on a slide in Vectashield (Vector Laboratories). The brains and VNCs were imaged with the same Olympus confocal microscope as legs and image stacks were analyzed in FIJI[61].

**Reconstruction and identification of hair plate axons in FANC EM dataset**
**Hair plate axon reconstruction and connectivity.** Hair plate axons were reconstructed in the electron microscopy dataset of the female ventral nerve cord, FANC[10,14], through manual proofreading of the automatically segmented cell fragments in Neuroglancer[62]. Annotations containing information about the reconstructed hair plate axons, such as what hair plate they project from, we uploaded to the Connectome Annotation Versioning Engine (CAVE)[63] and are publicly

available to those that have access to the FANC dataset. We then determined and proofread the cell fragments that received or provided at least 3 synapses from or onto hair plate axons. In addition to proofreading the upstream and downstream partners of hair plates, we classified them, if possible, into general cell types and specific cell classes using pre-existing CAVE annotation tables and/or expert knowledge[10,46,64,65]. Note, we classified glutamatergic neurons as inhibitory[66]. Our neuronal annotations were uploaded to CAVE and are publicly available. We determined the downstream and upstream partners of hair plates and performed connectivity analyses using a custom Python script. Note that only neurons that had 4 or more output synapses on average per neuron within a cell class onto individual downstream neurons were used in the reflex circuit analyses.

**Hair plate identification.** Hair axons were identified from previously traced backbones and annotations in CATMAID[14,42]. The axons of CxHP8, CxHP3, and CxHP4 were identified based on if they projected through the ventral prothoracic, dorsal prothoracic, or prothoracic accessory nerves, respectively (Supplementary Fig. S5A). All trochanter hair plate axons project through the prothoracic leg nerve (Supplementary Fig. S5A), so TrHP5, TrHP6, and TrHP7 neurons were instead identified based on their location in the nerve bundle, axonal morphology, and downstream connectivity with motor neurons (Fig. 5D). TrHP6 axons were identified because they make distinct connections onto motor modules compared to TrHP5 and TrHP7 (Fig. 5D). TrHP5 and TrHP7 axons could be differentiated because they enter the leg neuropil through the ProLN at different locations (Supplementary Fig. S5A).

**Motor impact score.** Motor impact score[21] was computed by first calculating the monosynaptic weight between a hair plate and motor

module, which is the total number of synapses that all neurons within a hair plate make onto the motor neurons within a motor module, divided by the total number of synapses onto that motor module across all neurons. Next, we calculated the di-synaptic weight between a hair plate and motor module by finding and summing the proportion of hair plate input synapses provided to each premotor neuron connected to the motor module of interest, multiplied by the proportion of input synapses those premotor neurons make onto the motor module. Note that the fractional input between these premotor neurons and the motor module is signed, based on the hemilineages of the premotor neurons. The motor impact score is then derived by summing the monosynaptic and di-synaptic weights.

### Calcium imaging of hair plate axons

The following preparation, setup, and analysis pipeline[25] was used to image calcium signals in hair plate axons in the neuromere of the left front leg during active and passive leg movements.

**Preparation.** We attached a fly to a custom-made holder, removed a rectangular piece of cuticle from the dorsal thorax, removed the underlying indirect flight muscles, and used an insect pin to displace the digestive system. This provided a dorsal view on the hair plate axons entering the VNC from the left front leg.

**Two-photon image acquisition.** Calcium signals in hair plate axons were recorded with a two-photon Movable Objective Microscope (MOM; Sutter Instruments) with a 20x water-immersion objective (Olympus XLUMPlanFI, 0.95 NA, 2.0 mm wd; Olympus). Hair plate neurons expressed the calcium indicator GCaMP7f (green fluorescence) and the structural marker tdTomato (red fluorescence). Fluorophores were excited at 920 nm by a mode-locked Ti:sapphire laser (Chameleon Vision S; Coherent). We used a Pockels cell to keep the power at the back aperture of the objective below ~35 mW. Emitted fluorescence was directed to two high-sensitivity GaAsP photomultiplier tubes (Hamamatsu Photonics) through a 705 nm edge dichroic beamsplitter followed by a 580 nm edge image-splitting dichroic beamsplitter (Semrock). Fluorescence was band-passed filtered by either a 525/50 (green) or 641/75 (red) emission filter (Semrock). Image acquisition was controlled with ScanImage 5.2 (Vidrio Technologies) in Matlab (MathWorks). The microscope was equipped with a galvo-resonant scanner, and the objective was mounted onto a piezo actuator (Physik Instrumente; digital piezo controller E-709). We acquired volumes of three 512 × 512 pixel images spaced 5 μm apart in depth (10 μm total) at a speed of 8.26 volumes per second.

**Two-photon image analysis.** Two-photon images were smoothed with a Gaussian filter (sigma = 3 pixels; size = 5 × 5 pixels). Each tdTomato image was aligned to the average tdTomato signal of the recorded trial using a cross-correlation-based image registration algorithm[67]. The same alignment was used for the GCaMP images. We averaged the three GCaMP and tdTomato images per volume. Then, we extracted the mean fluorescence in manually drawn regions of interest (ROIs). To correct for vertical movement of the VNC, we computed the ratio of GCaMP fluorescence to tdTomato fluorescence in each frame. To facilitate comparisons across trials and flies, ratio values were z-scored by subtracting the mean of a baseline ratio and dividing by the standard deviation of that baseline ratio. The baseline was defined in each trial as the 10% smallest ratio values. Finally, z-scored ratio values were upsampled to the sampling rate of leg tracking (300 Hz) using cubic spline interpolation and then low-pass filtered using a moving average filter with a time window of 0.2 s.

**Platform and treadmill.** The platform consisted of a metal pin (0.5 mm diameter, 4.4 mm length) mounted onto a three-axis micromanipulator (MP-225; Sutter Instruments). The pin was wrapped in

black sandpaper to provide sufficient grip for the flies' tarsi. The micromanipulator was controlled manually.

The treadmill consisted of a patterned Styrofoam ball (9.1 mm diameter; 0.12 g) floating on air in an aluminum holder. The air flow was set to ~500 ml/min. The ball was illuminated by two infrared LEDs (850–nm peak wavelength; ThorLabs) via optical fibers. Ball movements were recorded at 30 Hz with a camera (Basler acA1300-200um; Basler AG) equipped with a macro zoom lens (Computar MLM3X-MP; Edmund Optics). Ball rotations around the fly's cardinal body axes (forward, rotational, sideward) were reconstructed offline using FicTrac[68]. Rotational velocities of the fly were calculated based on the known diameter of the ball. Velocities were upsampled to the sampling rate of leg tracking (300 Hz) using cubic spline interpolation and low-pass filtered using a moving average filter with a time window of 0.2 s.

**Leg tracking.** Movement of the left front leg was recorded at 300 Hz with two cameras (Basler acA800-510um; Basler AG) equipped with 1.0x InfiniStix lenses (68 mm wd; Infinity) and 875 nm short pass filters (Edmund Optics). The leg was illuminated by an infrared LED (850–nm peak wavelength; ThorLabs) via an optical fiber. We used a previously trained[25] deep neural network[44] to automatically track all leg joints in each camera view. 2D tracking data from both camera views were then combined to reconstruct leg joint positions and angles in 3D using Anipose[45].

**Behavior classification.** Fly behavior was classified semi-automatically based on thresholds on the velocity of the front and middle leg tarsi as described previously[25]. Flies on the platform were classified as resting or actively moving. Flies on the treadmill were classified as resting, walking, or grooming. All classifications were reviewed and manually corrected if necessary.

**Data selection.** Frames were manually excluded from the analysis if the front leg was involved in movements other than walking or grooming on the treadmill (e.g., extended downward pushing), the femur-tibia joint of the front leg was not tracked correctly, or the two-photon image registration failed (e.g., the VNC moved out of the imaging volume).

### Optogenetic activation and silencing experiments

Optogenetic experiments were performed on adult male flies that were raised on 35 mM in 95% EtOH ATR for 1–3 days, were 2–5 days old, de-winged, and fixed to a rigid tether (0.1 mm thin tungsten rod) with UV glue (KOA 300). These flies were placed onto a spherical foam ball (weight: 0.13 g; diameter: 9.08 mm) suspended by air within a visual arena. A dark bar with a width of 30 degrees with respect to the fly oscillated at 2.7 Hz in front of the fly. A spatially precise red (638 nm; 1200 Hz pulse rate; 30% duty cycle, Laserland) or green (532 nm; 1200 Hz pulse rate; 60% duty cycle, Laserland) laser was then positioned on the thorax-coxa joint of the left front leg. Optogenetic activation experiments were conducted on flies in which their CxHP8 neurons expressed ChrimsonR, whereas silencing experiments were performed on those that expressed GtACR1 in those neurons (Table 1). Trials were 2 s in duration and consisted of turning the laser on (experimental trials) or keeping it off (control trials) 0.5 seconds into the trial for 1 second. During each trial, the behavior each fly was recorded with 6 high-speed cameras (300 fps; Basler acA800-510 μm; Balser AG) and the movement of the ball was recorded at 30 fps with a camera (FMVU-03MTM-CS) and processed using FicTrac[68]. The 3D positions of each leg joint and the corresponding joint angles were determined by using DeepLabCut[44] and Anipose[45]. Kinematic analyses were performed in a custom Python script. Behaviors were classified using a random forest classifier[45]. In addition, front leg grooming was also classified based on the proximity and velocity of the front legs.

## Treadmill experiments

Treadmill experiments were performed on 2-5 day old CxHP8 silenced and genetically-matched control male flies (Table 1) using an actuated treadmill system[35]. The inward rectifying potassium channel, Kir 2.1, was used to silence the activity of CxHP8 neurons throughout development. In short, trials consisted of placing a de-winged fly into a chamber and moving the belts of the split-belt treadmill at a steady-state speed of 10 mm/s for 20 minutes in total. During which, we recorded the behavior of the fly at 200 fps and in 20 second bouts with 5 high-speed cameras (Basler acA800-510 μm; Balser AG). We then used DeepLabCut[44] and Anipose[45] to reconstruct the 3D positions of head, thorax, abdomen, and each leg tip. Custom Python scripts were used to analyze and visualize walking kinematics and posture.

## Kinematic classification and parameters

The classification and quantification of behavior, swing and stance, and most walking kinematic parameters for tethered flies and flies walking on the treadmill were conducted in the same manner as a prior study[35]. For tethered flies in both the calcium imaging and optogenetic experiments, steps or frames were classified as forward walking if the instantaneous forward velocity and absolute rotational velocity of the ball was greater than 5 mm/s and less than 25 degrees/s, respectively. Left turns were classified as periods where the rotational velocity was less than −25 degrees/s and greater than −50 degrees/s. Right turns had a rotational velocity greater than 25 degrees/s and less than 50 degrees/s. Other behaviors, like rest/standing and front leg grooming, were determined by using a behavioral classifier[45]. For flies walking on the treadmill, forward walking was classified as periods where the body velocity had a forward velocity greater than 5 mm/s and the absolute heading angle with respect to the front of the chamber was less than 15 degrees. Rest periods are classified as those where the instantaneous velocity of the body and all tarsi was less than 5 mm/s (i.e., all legs in stance).

During walking, the classification of each leg into swing and stance was based on the smoothed (i.e., with a Gaussian Kernel) instantaneous velocity of the tarsus of each leg. Instantaneous values were negative during periods where the tarsus moved posteriorly along the longitudinal axis of the body. A leg was classified as being in swing if its tarsus had an instantaneous velocity greater than 5 mm/s or less than −25 mm/s. Otherwise, the leg was classified as being in stance. Short, 1 video frame, classifications of swing or stance were corrected and reclassified as the other phase of the step cycle.

Classified forward walking steps were further filtered to remove steps that occurred during the transition into forward walking or produced kinematics that were inconsistent with what was previously published[34,69–71]. Any step that didn't meet the following criteria for a forward walking step was removed: step frequency between 5 and 20 steps/s, a swing duration between 15 and 75 ms, and a stance duration less than 200 ms. Steps that passed these filtering criteria were used for further kinematic analyses and comparisons.

All kinematic analyses and visualizations were done in custom Python scripts, which are publicly available on GitHub (https://github.com/Prattbuw/Hair_Plate_Paper/releases/tag/v1.0.0).

## Statistical analyses

Linear mixed-effects models, which account for repeated measurements within flies, were used to statistically compare joint angle or tarsi position distributions in walking (at the swing-to-stance transition) and grooming flies during green laser presentation (optogenetic inactivation) and absence. These models were also used to statistically evaluate the anterior and posterior extreme positions and resting leg positions between control flies and flies with CxHP8 neurons chronically silenced. In these models, the fixed effects were the laser state or experimental group (predictor/independent variable) and the joint angles or positions, or leg positions (dependent variable). The random effect was fly identity, which accounted for the repeated and non-independent measures within flies, and common-group effects. Note that the linear mixed-effects models were implemented using the statmodels Python library. t-tests were used to compare posture and step kinematics between CxHP8 chronically silenced and control flies. A Kuiper two-sample test was used to assess statistical significance for inter-leg coordination. If multiple comparisons were performed, a Bonferroni correction was employed to adjust the threshold of significance.

## Reporting summary

Further information on research design is available in the Nature Portfolio Reporting Summary linked to this article.

## Data availability

The calcium imaging, optogenetic, and treadmill kinematic silencing datasets generated in this study have been deposited in the Dryad database under https://doi.org/10.5061/dryad.fxpnvx153. All datasets are publicly available. FANC connectome data were analyzed from the CAVE materialization version 840, timestamp 2024-01-17T08:10:01.179472. FlyWire connectome data were analyzed from the public release version 783. Any additional information required to reanalyze the data is available from the lead contact upon request.

## Code availability

Code for analyzing and visualizing hair plate connectivity in the EM dataset, calcium activity of CxHP8 neurons, kinematics during optogenetic experiments, and walking kinematics and posture during treadmill experiments is located on GitHub (https://github.com/Prattbuw/Hair_Plate_Paper/releases/tag/v1.0.0).

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

## Acknowledgements

We thank members of the Tuthill and Brunton Labs for technical assistance and feedback on the manuscript. We thank Seok Jun Moon for sharing the CxHP8 split-GAL4 driver line (R48A07 AD: R20C06 DBD). B.G.P. was supported by an NSF Graduate Research Fellowship (Fellow ID: 2018261272). C.J.D. was supported from the Deutsche Forschungsgemeinschaft (DFG, German Research Foundation) project 432196121. Other support was provided by National Institutes of Health grants R01NS102333, R01NS128785, and U19NS104655, a Searle Scholar Award, a Klingenstein-Simons Fellowship, a Pew Biomedical Scholar Award, a McKnight Scholar Award, a Sloan Research Fellowship, the New York Stem Cell Foundation, and a UW Innovation Award to J.C.T. J.C.T. is a New York Stem Cell Foundation – Robertson Investigator.

## Author contributions

B.G.P. and J.C.T. conceived the study and wrote the manuscript. I.S. created the gorgeous Blender model of a fruit fly with hair plates, as well as Supplementary Video 2 B.G.P. and A.S. performed confocal imaging of hair plates. B.G.P. and A.C. proofread the hair plate axons and their downstream and upstream partners in FANC. C.J.D. collected and processed the calcium imaging data for CxHP8. G.M.C. and S.W.B. collected optogenetic activation and silencing datasets for CxHP8. B.G.P. analyzed the connectivity, calcium imaging, and optogenetic datasets. B.G.P. collected and analyzed the kinematics and posture of flies walking in the treadmill setup. A.A. provided valuable feedback about the connectivity of hair plates onto motor circuits.

## Competing interests

The authors declare no competing interests.
