## [Transparent Peer Review file · Nature Communications]

Proprioceptive limit detectors contribute to sensorimotor control of the *Drosophila* leg

Corresponding Author: Dr John Tuthill

Version 1:

Reviewer comments:

Reviewer #4

(Remarks to the Author)

Pratt et al aim at investigating the role of hair plates, known as limit detectors of limb movements in insects. Using a combination of behavior tracking, genetic manipulation, physiological recording, they characterize a specific class of hair plates on the fly coxa, CxHP8, which is recruited during extreme positions of leg movements that are potentially necessary for behaviors like grooming. They also use connectomics to predict the involvement of other hair plates based on their morphology and downstream circuits. Future investigations will be needed to verify those predictions by characterizing the different hair plate types and their efferent circuitry using specific genetic reagents.

While more is known about proprioceptors encoding limb position and movement, this study is the first looking at this class of proprioceptors in *Drosophila*, establishing a new foundation for studying motor control in the fly. I think this merits publication even if hair-plates have been examined in other insects, given the genetic and connectomic toolkit of *Drosophila* will allow building on these results. The paper is well written, the figures are clear and the methodology/results are sound. In addition, the authors have addressed most of the points raised by reviewers in the first round. Here are a few more comments/suggestions, although I think the paper is more or less ready for publication as is.

Comments/suggestions:

- I understand that most other genetic lines are not as specific as the CxHP8 line, which limits the ability to precisely investigate the role of other hair plates. However, Table S1 indicates a line (VT038115AD:VT061711DBD) that covers TrHP5 and TrHP7, both predicted to be involved in posterior movement in Figure 5F. An experimental set consisting in manipulating these hair plates during walking/resting using this line might strengthen the point of the paper, which is to verify connectome-based predictions of hair plates functions.
- In addition, using broader lines presented in this paper might address if combinatorial manipulation of multiple hair plates leads to more drastic effect on locomotion. In a way, this could address the author statements lines 354/55.
- In Figure 3G, silencing of CxHP8 does not change the lateral position of the tarsus (only vertical), while it does in Figure 4C-D. Do the authors think those differences are potentially due to different behavioral paradigm (ball vs treadmill) and/or genetic silencers (GtACR1 vs Kir2.1)?
- In Figure 3D, Figure 4C-D, does the change in tarsus position during CxHP8 manipulation is a consequence of the effect on thorax-coxa joint, or rather an indirect effect on tarsus motor module (although Figure 2B, D show weaker connectivity from CxHP8 to tarsus pre-MNs/MNs).
- In Figure S1D, the activity of CxHP8 correlates with front leg grooming. It seems there is some variability in calcium signal/kinetics despite the similar duration of grooming? Also, shall we expect different (potentially higher) calcium signal during head grooming since the position of the front legs might reach even more extreme positions?
- In Figure 4, since the treadmill can be used to modulate speed, can we expect more drastic effects of CxHP8 silencing during faster walking bouts (i.e. >10mm/s)? Also, it would be interesting to see effect of broad optogenetic activation of CxHP8 in such paradigm.

Reviewer #5

(Remarks to the Author)

Responses to reviewer comments on “Proprioceptive limit detectors contribute to sensorimotor control of the *Drosophila* leg”

(Reviewer comments in black, our responses in red).

Reviewer #4:

Pratt et al aim at investigating the role of hair plates, known as limit detectors of limb movements in insects. Using a combination of behavior tracking, genetic manipulation, physiological recording, they characterize a specific class of hair plates on the fly coxa, CxHP8, which is recruited during extreme positions of leg movements that are potentially necessary for behaviors like grooming. They also use connectomics to predict the involvement of other hair plates based on their morphology and downstream circuits. Future investigations will be needed to verify those predictions by characterizing the different hair plate types and their efferent circuitry using specific genetic reagents.

While more is known about proprioceptors encoding limb position and movement, this study is the first looking at this class of proprioceptors in *Drosophila*, establishing a new foundation for studying motor control in the fly. I think this merits publication even if hair-plates have been examined in other insects, given the genetic and connectomic toolkit of *Drosophila* will allow building on these results. The paper is well written, the figures are clear and the methodology/results are sounds. In addition, the authors have addressed most of the points raised by reviewers in the first round. Here are a few more comments/suggestions, although I think the paper is more or less ready for publication as is.

Comments/suggestions:

I understand that most other genetic lines are not as specific as the CxHP8 line, which limits the ability to precisely investigate the role of other hair plates. However, Table S1 indicates a line (VT038115AD:VT061711DBD) that covers TrHP5 and TrHP7, both predicted to be involved in posterior movement in Figure 5F. An experimental set consisting in manipulating these hair plates during walking/resting using this line might strengthen the point of the paper, which is to verify connectome-based predictions of hair plates functions.

We appreciate the suggestion and have previously considered this approach of using lines that label neurons of just two hair plates to infer their function during various behaviors. However, we have two concerns with such an approach. First, prior work in stick insects showed that ablating additional hair plates on the middle leg had non-linear effects on the positioning of the ipsilateral hind leg (Cruse et al., 1984, *J. Comp. Physiol. A*; Fig. 3). Therefore, the relative contribution of a single hair plate to the control of the leg may be difficult to gleam from manipulations of multiple hair plates, such as TrHP5 and TrHP7. Second, VT038115AD:VT061711DBD and other hair plate driver lines we made label only a subset of the neurons of each hair plate (i.e. 2 neurons of TrHP5 and 2 neurons of TrHP7). Given this incomplete labeling, it would be challenging to dissect the relative effect of the 2 cells from each hair plate on leg motor control. Having said all that, we think

the approach you proposed may be feasible in the case where driver lines have nearly complete labeling of neurons with each hair plate, and when there are lines that label neurons of one unique and one overlapping hair plate, which would enable pairwise assessments of hair plate function. We have added this potential approach to revealing hair plate sensorimotor function to the discussion (lines: 283-285).

In addition, using broader lines presented in this paper might address if combinatorial manipulation of multiple hair plates leads to more drastic effect on locomotion. In a way, this could address the author statements lines 354/55.

Thank you for the suggestion. In a previous study (Pratt et al., 2024, *Current Biology*; Fig. S2), we showed that chronically silencing subsets of CxHP4, TrHP5, TrHP6, TrHP7, and TrCS3 neurons in addition to a subset of CxHP8 neurons resulted in impaired step kinematics and inter-leg coordination, particularly at faster walking speeds. Therefore, manipulations of multiple hair plates do result in more drastic effects on locomotion. Recognizing that this finding would be useful to mention in this study, we have now included it to provide support for the statements in now lines 356-357 (lines: 268-270).

In Figure 3G, silencing of CxHP8 does not change the lateral position of the tarsus (only vertical), while it does in Figure 4C-D. Do the authors think those differences are potentially due to different behavioral paradigm (ball vs treadmill) and/or genetic silencers (GtACR1 vs Kir2.1)?

Thank you for pointing out this key difference in tarsus position between the ball and treadmill setups. We think that the difference in tarsus position is the result of both the behavioral paradigm and silencing method. We previously showed that step kinematics and inter-leg coordination are different between tethered flies walking on the ball and untethered flies walking freely or on the treadmill, which we attributed to differences in substrate (i.e. flat versus spherical) and the fact that tethered flies don't bear their weight (Pratt et al., 2024, *Current Biology*; Fig. 2-3). In regard to genetic silencing, Kir 2.1 is expressed throughout development, silencing CxHP8 neurons for the life of the fly, which may lead to compensatory effects on kinematics. On the contrary, GtACR1 allows for temporally precise control over the inhibition of CxHP8 neurons, likely mitigating the chance of cellular or circuit compensation. In fact, significant changes in extreme leg positioning only became apparent 3 days after the hairs of a hair plate were chronically deflected with plasticine and hair rows removed in a stick insect (Bässler, 1977, *Biol. Cybernetic*), suggesting that compensatory effects can occur after hair plate manipulations. We have now added a sentence to the results (lines: 186-189) that describes our thoughts on the difference in tarsus position.

In Figure 3D, Figure 4C-D, does the change in tarsus position during CxHP8 manipulation is a consequence of the effect on thorax-coxa joint, or rather an indirect effect on tarsus motor module (although Figure 2B, D show weaker connectivity from CxHP8 to tarsus pre-MNs/MNs)?

Thank you for the insightful question. The change in tarsus position after CxHP8 manipulations is most likely a consequence of impaired thorax-coxa joint movement through altered coxa posterior motor module activity. First, our motor impact analysis (Fig. 5E) showed that when considering a 2-layered network (i.e. all direct and disynaptic connections) that CxHP8 connectivity has very little impact on tarsus related motor modules, but very strong excitatory impact on the coxa posterior motor module. Second, our optogenetic activation experiments in standing flies (Fig. 3E-D) showed that activating CxHP8 neurons resulted in thorax-coxa movement that supported the corresponding changes in tarsus movement. However, to your point, sensorimotor networks are more complex than what we analyzed here, having state-dependent dynamics and recurrent connections, so we can't rule out the possibility that some of the tarsus movement we observed after CxHP8 manipulations was due to tarsus motor modules. We mention this possibility of leg-wide control rather than leg segment only control of hair plates in the Discussion (lines: 246-250).

In Figure S1D, the activity of CxHP8 correlates with front leg grooming. It seems there is some variability in calcium signal/kinetics despite the similar duration of grooming? Also, shall we expect different (potentially higher) calcium signal during head grooming since the position of the front legs might reach even more extreme positions?

Thank you for your keen observation. Yes, calcium signals during different front leg grooming bouts vary, likely because of the exact kinematic nature of the grooming bout and its consequence on the number and degree to which CxHP8 cuticular hairs are deflected. In this study, we were coarse in our classification and kinematic characterization of front leg grooming, but given a more granular analysis, we suspect the variability in calcium activity would be explained by differences in how the fly was grooming their front legs. As for calcium signals during head grooming, we expect they would also be variable on a grooming bout-by-bout basis, but possibly different in both their kinetics and magnitude than front leg grooming because of the inherently different limb articulations and kinematics (e.g. Ravbar et al., 2021, *eLife* and Lobato-Rios et al., 2022, *Nature Methods*). A detailed 3D kinematic description of joint angle distributions would be valuable in determining how CxHP8 calcium signals map onto different types of grooming. We now provide an explanation of the calcium signal variability in the Results (lines: 97-99).

In Figure 4, since the treadmill can be used to modulate speed, can we expect more drastic effects of CxHP8 silencing during faster walking bouts (i.e. >10mm/s)? Also, it would be interesting to see effect of broad optogenetic activation of CxHP8 in such paradigm.

For this study, we collected data of flies with CxHP8 neurons silenced walking on the treadmill at one driving speed (i.e. 10 mm/s) to simplify comparisons. However, as mentioned in the second comment above, we previously found that flies with a swath of proprioceptors silenced, including a subset of CxHP8 neurons, had impaired step kinematics and inter-leg coordination, especially as

walking speed increased (Pratt et al., 2024, *Current Biology*; Fig. S2). Therefore, we would expect a similar finding for flies that just had CxHP8 neurons silenced.

We agree, it would be interesting to see the effect of broad optogenetic activation of CxHP8 neurons in flies walking on the treadmill. Nevertheless, we steered away from this direction in this study because we find it difficult to interpret the impact of injecting mistimed sensory feedback related signals into sensorimotor circuits in walking flies. Specifically, it is unclear to us how activating CxHP8 at times when they would not be active during the step cycle would inform us about how CxHP8 feedback is used to control leg movement compared to using silencing methods. Our expectation given the optogenetic results shown in Fig. 3 would be that optogenetic activation would result in posterior leg movement in regions of the step cycle where anterior drive isn't as strong, presumably the end of swing and during the stance phase. An interesting future experiment would be to precisely and transiently activate CxHP8 neurons during a specific phase of the step cycle using closed-loop methods and determine how the fly compensates for such a perturbation.

Reviewer #5:

Thank you for co-reviewing this manuscript. We appreciate and value your feedback.